# Acetoin Promotes Plant Growth and Alleviates Saline Stress by Activating Metabolic Pathways in Lettuce Seedlings

**DOI:** 10.3390/plants13233312

**Published:** 2024-11-26

**Authors:** Chaowei Zhou, Hui Shen, Shangbo Yan, Changyi Ma, Jing Leng, Yu Song, Nan Gao

**Affiliations:** 1School of Biotechnology and Pharmaceutical Engineering, Nanjing Tech University, Nanjing 211816, China; 202162118032@njtech.edu.cn (C.Z.); shenhui0410@163.com (H.S.);; 2School of 2011, Nanjing Tech University, Nanjing 211816, China

**Keywords:** bio-fertilizer, transcriptome, volatile organic compounds, saline stress, photosynthesis

## Abstract

Acetoin is a volatile organic compound, which is a class of metabolites produced by plant growth-promoting rhizobacteria. The mechanisms underlying plant growth promotion by acetoin and its potential to induce saline stress tolerance in plants are poorly understood. Lettuce (*Lactuca sativa* L. var. ramosa Hort.) seedlings in hydronics and pots under non-saline or saline conditions were foliar-sprayed with 10 mL of 0 or 1 mg·mL^−1^ acetoin at 7 and 14 d after transplantation and harvested 7 d after the second spray. Shoots and roots of hydroponic lettuce seedlings were harvested at 6 and 24 h after treatment for RNA sequencing. Seedlings sprayed with acetoin showed more vigorous growth, with higher shoot and root biomass than those of the controls, in both hydronic and pot modes. The transcriptomic analysis revealed acetoin application resulted in 177 differentially expressed genes (39 upregulated and 138 downregulated) in shoots and 397 differentially expressed genes (112 upregulated and 285 downregulated) in roots. These DEGs, mainly involved in plant hormone signal transduction and the mitogen-activated protein kinase, have the potential to trigger plants’ responses to various environmental stimuli, including stress and developmental signals. Under saline conditions, acetoin-treated plants showed increased net leaf photosynthesis and activities of several defense enzymes, indicating that acetoin enhances both fundamental growth and the plant’s stress defenses, especially against salinity. In summary, acetoin appears to act through a complex interplay of genetic and biochemical mechanisms, influencing key signaling pathways and physiological processes that lead to improved growth and stress tolerance in lettuce seedlings.

## 1. Introduction

Chemical fertilizer application can ensure high crop yield; however, the long-term application of chemical fertilizers can affect the biological and physicochemical health of the soil ecosystem, crop quality, and environment, leading to a decline in agricultural productivity [1,2]. The decreased productivity of crop plants is also influenced by varied environmental factors, such as heat, drought, and salinity [2,3,4]. Among those factors, an increase in soil salinity is one of the main challenges of contemporary agriculture. The global area of salt-affected soils is approximately 1.0×10^9^ ha, accounting for approximately 25% of Earth’s land area [5]. Human activities such as irrigation with saline water, deforestation, and improper agricultural practices involving the excessive use of chemical fertilizers exacerbate soil salinity [6]. Hence, biofertilizers have attracted extensive attention owing to their ability to protect the environment, enhance soil fertility, and improve crop productivity.

As an eco-friendly alternative to chemical fertilizers, plant growth-promoting rhizobacteria (PGPR) are used widely to influence overall plant health by contributing to supporting nutrient acquisition of the host plant and enhancing resistance to biotic and abiotic stress [7]. PGPR employs several mechanisms to provide benefits to crops, including atmospheric nitrogen fixation; solubilization of soil nutrients such as phosphate, zinc, and potassium; and secretion of plant growth-promoting substances, for instance, auxins, ethylene, gibberellins, and volatile organic compounds (VOCs) [8,9,10]. Despite their benefits, the compounds secreted by PGPR are changeable and unstable regarding dynamic environmental factors, which can lead to a failure in maintaining their effectiveness at a consistently positive level.

VOCs, which have a low molecular weight, weak polarity, low boiling point, and high lipophilicity, have been shown to play various roles, including acting as inter- and intra-species signal compounds, stimulating and inhibiting plant growth, and affecting phytopathogens [9,10]. Additionally, VOCs have the potential to promote plant growth and seed germination, increase nutrient uptake [11,12], induce systemic resistance in crops [13], and block the growth of pathogenic bacteria [10,14].

Lettuce (*Lactuca sativa* L. var. ramosa Hort.), revered for its rich nutritional profile and diverse health benefits, is one of the most important vegetables worldwide, but its moderate sensitivity to salinity seriously restricts yield [4,15]. Studies have proved that its plant physiology is impaired by the osmotic stress response to salinity, resulting in stomatal closure to decrease water loss by transpiration [16]. Furthermore, photosynthesis is affected by Na^+^ toxicity, where a high cytosolic accumulation of Na^+^ tends to replace some K^+^, and consequently, vital enzymatic activities involving K^+^ are disrupted [17]. Additionally, salinity has been demonstrated to induce the modulation of endogenous phytohormones levels, which in turn affect the signaling pathways (e.g., mitogen-activated protein kinase [MAPK] signaling pathway) involved in the downstream changes in roots, leaves, and cellular structures [18].

Acetoin is a VOC produced by PGPR. Acetoin promotes plant growth, helps prevent plant diseases [19], and enhances plant resistance to biotic and abiotic stressors [9]. However, according to our review of the literature, the effects of exogenous acetoin on any global transcriptome changes in plants have not been investigated. In this study, we investigated the effects of acetoin on lettuce growth under different cultivation modes, including on plants under salt stress. RNA sequencing (RNA-seq) was used to explore the promotion or potential effects of stress tolerance in plants treated with acetoin. We hypothesized that (1) acetoin promotes the growth of lettuce plants by increasing the photosynthetic capacity and expression of functional genes related to plant growth, and (2) acetoin has the potential to induce stress tolerance in lettuce plants by enhancing defense enzyme activities and the expression of defense-related signaling genes. This study presents evidence for acetoin-induced transcriptomic changes in lettuce and their potential to induce stress tolerance.

## 2. Results

### 2.1. Phenotypic Response of Hydroponic Lettuce to Acetoin

For identification of the phenotype of lettuce seedlings sprayed by acetoin, the lettuce seedlings were cultured in a hydroponic solution and sprayed with different concentrations of acetoin (0, 0.5, 1, and 2 mg·mL^−1^) in the hydroponic experiment. Foliar application of acetoin significantly increased the biomass accumulation and root development of lettuce seedlings. Compared to the control treatment (0 mg·mL^−1^), there was an obvious apparent elongation of roots among the different acetoin-treated seedlings (Figure 1A). The dry weights of shoot and root were significantly increased by the application of 1 mg·mL^−1^ acetoin, exhibiting 18.08 and 68.68% increases compared to those of the control, respectively (Figure 1B). Both low (0.5 mg·mL^−1^) and high (2 mg·mL^−1^) concentrations of acetoin exhibited similar effects. Acetoin in any concentration had no obvious impact on the number of leaves (Figure 1B). However, the leaf-greenness index of seedlings was enhanced with the low concentration of acetoin, and there were no significant changes with higher concentrations (Figure 1D).

### 2.2. RNA Sequencing and Analysis of DEGs

In total, 24 samples were used to analyze dynamic gene expression changes in lettuce seedlings after acetoin spraying in the hydroponic experiment. After filtering out adaptor sequences and low-quality reads, 1,057,224,748 clean reads were obtained (Appendix A). The Q20 and Q30 values of each sample were greater than 96% and 90%, respectively, and the GC content was approximately 45.6% (Appendix A). Subsequently, all clean reads were mapped to the reference genome; the ratio of mapped reads for each sample ranged from 92.32 to 96.23% (Appendix A).

Per DEG analysis, 544 genes were differentially expressed, of which 151 were upregulated, and 393 were downregulated (Appendix A). At 6 and 24 h post acetoin treatment, 47 and 130 DEGs were, respectively, observed in the leaves, along with 304 and 93 DEGs in the roots, compared with those in the control group (Appendix A).

### 2.3. GO Classification Analysis of DEGs

The 544 identified DEGs were divided into biological process, cell component, and molecular function ontologies, and each DEG was assigned at least one GO term. A total of 31 significantly enriched level-2 GO terms were obtained from the 544 DEGs, including 17 biological processes, 10 cellular components, and 4 molecular functions (Figure 2). The largest portion of genes was associated with biological processes and cell components, with relatively few changes in the expression of genes related to molecular function (Figure 2).

### 2.4. KEGG Enrichment Pathway Analysis of DEGs

Among the identified 544 DEGs, 282 were annotated in the KEGG database and divided into 5 KEGG A classes, 15 KEGG B classes, and 68 pathways (Figure 3). In total, there were 12 significantly enriched pathways across all samples from 6 and 24 h post acetoin treatment (Q value ≤ 0.05; Table 1). In the shoot samples, at 6 h post acetoin treatment, genes were significantly enriched in flavonoid biosynthesis, plant hormone signal transduction, and MAPK signaling pathways; at 24 h post acetoin treatment, genes were significantly enriched in the MAPK signaling pathway–plant, plant hormone signal transduction, and plant–pathogen interaction pathways (Table 1). In root samples of plants 6 h post acetoin treatment, the pathways of significant enrichment were biosynthesis of secondary metabolites, phenylpropanoid biosynthesis, fatty acid degradation, metabolic pathways, tyrosine metabolism, and biosynthesis of unsaturated fatty acids compared to those of control samples (Table 1). Plant hormone signal transduction and MAPK signaling pathways involved in relieving saline stress in plants were selected for further analysis.

### 2.5. Comparison of DEGs Related to Plant Hormone Signal Transduction Pathways

Plant hormone systems play a crucial regulatory role in plant growth; the ethylene signal transduction pathway is one such pathway. Among the metabolic genes of the ethylene signal transduction pathway, the genes encoding EIN3-binding F-box protein 1 (*EBF1*) (ncbi_111917430, 1.5-fold) and EIN3-binding F-box protein 2 (*EBF2*) (ncbi_111921717, 1.2-fold) were downregulated at 6 h after acetoin treatment, and the genes encoding serine/threonine-protein kinase CTR1 (*CTR1*) (ncbi_111906571, 1.3-fold) and *EBF2* (ncbi_111921717, 1.1-fold) were downregulated at 24 h after acetoin treatment compared to that in controls (Figure 4 and Appendix A). In the ABA signaling pathway, the gene encoding ABA receptor PYR1-like protein (*PYL4*) (ncbi_111879857, 1.6-fold) was upregulated at 6 h after acetoin treatment, and the gene encoding protein phosphatases 2C (*PP2CA*) (ncbi_111910618, 1.0-fold) was downregulated at 24 h after acetoin treatment compared to those in controls (Figure 4 and Appendix A).

### 2.6. Comparison of DEGs Related to the MAPK Signaling Pathway–Plant

The MAPK signaling pathway is a key element in plant defense reactions. Genes such as *EBF1* (ncbi_111917430, 1.5-fold) and *EBF2* (ncbi_111921717, 1.2-fold) related to the ethylene signal transduction pathway were downregulated, and *PYL4* (ncbi_111879857, 1.6-fold), related to the ABA signaling pathway, was upregulated at 6 h after acetoin treatment (Figure 5 and Appendix A). *CTR1* (ncbi_111906571, 1.3-fold), mitogen-activated protein kinase kinase 9 (*MKK9*) (ncbi_111915142, 1.3-fold), and *EBF2* (ncbi_111921717, 1.1-fold) related to ethylene signal transduction were downregulated; *PP2CA* (ncbi_111910618, 1.0-fold) related to ABA signaling pathway was downregulated; and *CP1* (ncbi_111914172, 1.2-fold) and *CML46* (ncbi_111920042, 1.2-fold) encoding calmodulin (CALM) and calmodulin-like protein (CML) were downregulated at 24 h after acetoin treatment (Figure 5 and Appendix A).

### 2.7. Validation of RNA-seq Data by Using qRT-PCR

The amplification efficiency of all tested primers was between 90% and 105%, and the dissolution curves were all single peaks, indicating that these primers could be used for qRT-PCR. No significant differences were observed between the three internal reference genes across all treatments. The relative expression of eight of the genes was consistent with the expression levels obtained from RNA-seq, and two genes (*CML46* and *PP2CA*) showed the same expression tendencies (Figure 6 and Appendix A), indicating that the transcriptome sequencing results were reliable.

### 2.8. Effects of Acetoin on Growth and Alleviation of Saline Stress

Saline stress inhibited the growth of lettuce seedlings, and acetoin treatment promoted growth and relieved saline stress in lettuce seedlings in the pot experiment (Figure 7). Compared with those in CK group, the fresh and dry shoot weights of acetoin-treated lettuce seedlings increased significantly (*p* < 0.05) by 24.8% and 26.5%, respectively (Figure 7). Under saline conditions, plant height, fresh and dry shoot weights, and fresh root weight of lettuce seedlings in the acetoin group increased significantly (*p* < 0.05) by 15.5%, 29.0%, 26.5%, and 40.9%, respectively, compared to those in the CK group (Figure 2).

Compared to those of the control seedlings, Mg^2+^ concentrations in leaves of lettuce were significantly decreased by saline stress, but K^+^ and Na^+^ concentrations and the K^+^/Na^+^ ratio of lettuce seedlings were increased significantly. Under non-saline conditions, Ca^2+^ and Mg^2+^ concentrations in the roots were significantly (*p* < 0.05) increased in acetoin-treated seedlings compared to those in CK seedlings (Table 2). Under saline conditions, acetoin-treated seedlings showed significantly (*p* < 0.05) increased Ca^2+^ concentrations and significantly (*p* < 0.05) decreased Na^+^ concentrations compared to controls (Table 2).

### 2.9. Effects of Acetoin on Leaf Photosynthetic Index and Defense Enzyme Activity

Acetoin had a favorable effect on the photosynthetic competence of lettuce seedlings. Within 7 d, the Pn, E, and gsw of lettuce leaves in the NaCl group were significantly lower (*p* < 0.05) than those in the CK group (Figure 8A–C), and Ci was not significantly changed (*p* < 0.05) (Figure 8D. After acetoin treatment in saline conditions, the photosynthetic indices (Pn, E, and gsw) of seedlings were consistently higher than those of untreated seedlings (Figure 8A–C).

Under non-saline conditions, SOD and MDA activities and soluble protein content were not significantly different (*p* < 0.05) between the CK and CK+ acetoin groups (Figure 9). SOD activity significantly increased (*p* < 0.05) in NaCl lettuce leaves compared to that in CK seedlings. On days 1, 2, and 3 after acetoin treatment, SOD activity in the NaCl + acetoin group was significantly higher (*p* < 0.05) than that in the NaCl group (Figure 9A). Compared with that in the CK group, MDA activity significantly increased (*p* < 0.05) in the NaCl group and significantly decreased (*p* < 0.05) with the application of acetoin at similar levels as those in the CK and CK+ acetoin groups (Figure 9B). Soluble protein content significantly decreased (*p* < 0.05) in the NaCl group compared to that in the CK group and was restored after acetoin treatment (Figure 9C). POD and CAT activities across the groups were not significantly different (Appendix A).

## 3. Discussion

### 3.1. Acetoin Promotes Plant Growth and Alleviates Saline Stress

Acetoin is an active signaling molecule that promotes plant growth and stress resistance. Studies have shown that acetoin effectively promotes various growth parameters in lettuce [19]. In this study, we investigated the effects of acetoin on lettuce under hydroponic and soil conditions. We observed a significant increase in dry weights of lettuce shoots and roots due to acetoin (1 mg·mL^−1^) (Figure 1). Moreover, the descending growth parameters of seedlings under saline stress, such as plant height, shoot weight, and root weight, were significantly alleviated by acetoin (Figure 7). Thses findings are consistent with those of existing studies showing that acetoin can induce plants to develop systemic resistance to salinity and effectively inhibit external stress and promote plant growth [20]. Additionally, acetoin has potential to enhance the symbiotic relationship between microbes and plants [21]. However, further research should explore the dosage, usage time, and promotion mechanisms of acetoin on different crops on the basis of differences in physiological habits and cultivation environment.

Photosynthesis is the basis of plant growth [22]. Studies have shown that when plants are exposed to salt stress, photosynthesis levels decrease, hindering the normal growth of plants [23]. We found that the Pn, E, and Ci of lettuce decreased significantly under saline conditions. After applying acetoin, the photosynthetic index of lettuce significantly increased, potentially promoting favorable regulation of lettuce growth and nutrient accumulation (Figure 8A–C).

The defense response to environmental stress can also affect plant growth. Salt stress is one abiotic stressor affecting plant growth and yield, and the Na^+^/K^+^ ratio is a critical indicator to measure the degree of salt stress. Maintaining an appropriate Na^+^/K^+^ ratio is essential to alleviating salt stress [24]. The main effect of salt stress is that the constant accumulation of Na^+^ can lead to a decrease in K^+^ content. In our study, under saline conditions, the Na^+^ content of seedlings increased significantly. After acetoin treatment, the K^+^ shoot content significantly increased, significantly increasing the K^+^/Na^+^ ratio, effectively alleviating the harmful effects of salt stress (Table 2).

Salt stress in plants can also cause the accumulation of reactive oxygen species [25] such as hydrogen peroxide (H_2_O_2_) and superoxide free radicals (O^2−^), resulting in an imbalance of cellular active oxygen metabolism and affecting plant growth and development [23]. Plants can accelerate the removal of reactive oxygen species (ROS), repair damage, maintain active oxygen metabolism, and reduce damage to the plant caused by salt stress by increasing the activity of antioxidant enzymes such as SOD and POD [26]. Our results showed that under saline conditions, SOD activity significantly increased after the application of acetoin compared to in controls (Figure 9A). Furthermore, MDA activity was significantly (*p* < 0.05) higher, and the soluble protein content was significantly (*p* < 0.05) lower in the NaCl control group than in the CK group, and the contents of both were restored with the application of acetoin (Figure 9B,C). These results indicate that acetoin can alleviate the hazardous effects of ROS in lettuce and contribute to its growth and development.

We conducted transcriptome analysis to understand the growth mechanisms of plants under salt stress with acetoin treatment. RNA-sequencing technology is increasingly used to characterize genomes and transcriptomes and has considerably increased the speed and efficiency of gene discovery. This sequencing technology can be used to determine global gene expression differences between populations or species, phenotypes, and responses to environmental stress [27]. In this study, whole-transcriptome analysis of lettuce leaves and roots in response to acetoin treatment was performed using RNA-seq. In total, 544 DEGs were identified in lettuce leaves and roots at 6 and 24 h post acetoin treatment when compared to that in the control groups (Appendix A). The GO and KEGG pathway enrichments for identified DEGs were analyzed. Significant GO enrichment was observed for metabolic processes, cellular processes, single biological processes, catalytic activity, and binding activity (Figure 2). Additionally, the notable KEGG pathways enriched in acetoin-treated lettuce were “plant hormones and signal transduction” and “MAPK signaling pathway–plant” in leaves at 6 and 24 h (Table 1). Many genes related to growth and defense response were activated, indicating that acetoin may enhance plant growth and resistance to salinity.

### 3.2. Expression of Genes Related to Plant Hormones and Signal Transduction

Plant hormones control plant growth and almost all other relevant developmental processes. In addition to its role in plant growth and development, ethylene is involved in regulating plant defense responses to biotic and abiotic stressors [26]. *CTR1* functions downstream of the ethylene receptor as a negative regulator of ethylene signaling, and *EBF1* and *EBF2* degrade transcription factors related to the ethylene signaling pathway [28]. ERF1 is an upstream component of ABA signaling. Evidence suggests that ERF1 plays an active role in salt stress tolerance via ethylene signaling [29]. Our study showed that genes related to ethylene metabolism (*CTR1*, *EBF1*, and *EBF2*) were downregulated in the leaves at 6 and 24 h after acetoin treatment (Figure 3 and Appendix A). This possibly affected the transcriptional regulators ethylene-insensitive protein 3 (EIN3) and the downstream of ethylene-responsive transcription factor 1/2 (ERF1/2), completing the ethylene signal output. Decreased expression of ERF1 may have also affected ABA signaling, which plays an important role in plant growth and defense response.

ABA, a phytohormone, has been shown to play an essential role in regulating plant growth and response to abiotic stresses such as heat, drought, and salinity [8,30]. ABA receptors include regulatory components of ABA receptors, pyrabactin resistance 1, and PYR1-like proteins (PYLs) [30]. The PYL family of ABA receptors plays an active role in plant growth, development, and stress resistance [30], and PP2C acts as a negative regulator, impairing plant growth and activation of stress response during ABA metabolism. Acetoin treatment caused the overexpression of *PYL* in leaves after 6 h and decreased the expression of *PP2CA* in leaves after 24 h (Figure 3 and Appendix A); both responses may improve plant growth and stress adaptation. Moreover, the inhibited phosphatase activity of PP2C could lead to the release of SNF1-related type 2 protein kinase (SnRK2) from the PP2C-SnRK2 complex and increase the activity of SnRK2 phosphokinase [31]. SnRK2 phosphorylates downstream target proteins, further activating basic leucine zipper transcription factors such as ABA-response elements and ABA-response element binding factors [32].

### 3.3. Expression of Genes Related to MAPK Signaling Pathway–Plant

The MAPK cascade is a major downstream pathway in eukaryotes that converts extracellular stimuli into sensors and receptors for intracellular responses [33]. The MAPK cascade is a relevant signal transduction pathway involved in plant cell differentiation and development [34] and responses to various biotic and abiotic stressors, including drought, low temperature, salinity, and hormones.

The mitogen-activated protein kinase in the MAPK cascade is activated by the core ABA signaling pathway composed of PYL and RCAR-PP2C-SnRK2, contributing to the enhancement of stress resistance in crops [33]. The PYL and PP2C family proteins, members of the ABA signaling pathway, play several roles in the regulation of plant MAPK pathways and are effectively activated during the stress response [35]. With acetoin treatment, the activity of SnRK2 phosphokinase was enhanced by the upregulation of *PYL4* and downregulation of *PP2CA* (Figure 4 and Appendix A), which enabled plants to have the potential to resist stress. Moreover, CTR1, a unique MAPKKK and important negative regulator in ethylene signaling appears to control branching cascades and antagonize MAPK cascades that target the same key nuclear transcription factors in ethylene signaling, allowing plants to regulate and adapt to the external environment [36]. In this study, the expression of *CTR1* was downregulated in acetoin-treated seedlings (Figure 4 and Appendix A), which likely had a positive effect on defense response.

Additionally, the binding of CALM/CML to Ca^2+^ was shown to affect MAPK signal transduction [37], further activating mitogen-activated protein kinase 8 (MPK8) that negatively regulates ROS production by controlling the expression of respiratory burst oxidase homologous protein D (RbohD) [38]. ROS play a key role in plant development and adaptability to salt stress [39]. In our study, *CALM/CML* expression was downregulated in leaves after acetoin treatment (Figure 4 and Appendix A), which may have led to the upregulation of ROS signaling. Additionally, we demonstrated that the removal of ROS was possibly caused by the increasing activity of SOD under salt stress. In general, these regulatory genes induced by acetoin were associated with signal transduction under adversity stress, which indirectly implies its capacity to alleviate the effects of stressors such as heat, drought, and salinity.

## 4. Materials and Methods

### 4.1. Experimental Materials

Lettuce seeds were purchased from Woshu Seed Industry Co. Ltd, Nanjing, China. Acetoin was purchased from Sigma-Aldrich Chemical Company, USA. Hoagland nutrient solution was purchased from Qingdao Hope Bio-Technology Co., Ltd., China. Soil was collected from a long-term, traditional, open vegetable plot in Yixing, Jiangsu Province, China (N 30°12′23″, E 119°52′89″). The soil was air-dried and sieved through a 2 -mm mesh. The saline soil was obtained by adding 51 mM of NaCl, and then mixed thoroughly. The physicochemical characteristics of the soil were reported in existing research [40], as follows: 11.0 g kg^−1^ organic matter, 29.7 mg kg^−1^ alkali-hydrolyzed nitrogen, 68.2 mg kg^−1^ available P, 271.2 mg kg^−1^ available K, and pH 7.08.

### 4.2. Greenhouse Hydroponic and Pot Experiments

Hydroponic experiments were performed using four spray solutions (0, 0.5, 1, and 2 mg·mL^−1^ acetoin), conducted in a humidified plant growth chamber (MLR-352H; Panasonic Healthcare Co., Ltd., Oizumi-Machi, Japan) controlled at 25 ± 1 °C, with 70% relative humidity, under a 14/10 h (light/dark) photoperiod. The seeds were surface-sterilized by immersion in 2% (*w*/*v*) sodium hypochlorite for 5–10 min and then rinsed 8–10 times with sterile distilled water. Surface-sterilized seeds were germinated for 7 d in a hydroponic system containing one-fifth of Hoagland solution [41,42]. Uniform germinated seedlings were transferred to a new Hoagland’s hydroponic system and grown for 21 d. The lettuce seedlings were sprayed with acetoin at 7 and 14 d after transplantation. The Hoagland solution was changed every 2–3 d. The growth parameters of lettuce seedlings were determined at 28 d after sowing.

The pot experiments were performed using two soil types (0 or 3 g NaCl kg^−1^ soil) and two spray solutions (0 or 1 mg·mL^−1^ acetoin), conducted in a greenhouse controlled at 25 ± 2 °C, 50–70% relative humidity, with a 14 h/10 h (light/dark) photoperiod. Surface-sterilized seeds were sown and germinated in sieved soil. The uniform seedlings within two pairs of true leaves were selected and transplanted into 360 mL paper pots containing 320 g of fresh soil, with two seedlings in each pot. Each treatment group contained five pots, with two seedlings in one pot serving as a single replication. The lettuce seedlings were sprayed with 10 mL of 0 or 1 mg·mL^−1^ acetoin at 7 and 14 d after transplantation. The position of the paper pots within the greenhouse was changed daily to ensure randomization, and the seedlings were irrigated every two days. The seedlings were harvested 21 d after transplantation.

### 4.3. Growth Parameters and Nutrient Elements Analysis

The leaf chlorophyll content was measured using a SPAD-502 chlorophyll meter (Minolta Corporation, Ltd., Osaka, Japan). We randomly selected eight fully expanded leaves from each treatment group to ensure reliable results. These leaves were scanned using a Microtek ScanMaker i800 Plus system (Wseen, Hangzhou, China). Leaf area was calculated using LA-S Leaf Area Analysis software (Version number 2.2.3.8, Wseen, Hangzhou, China). Shoot and root fresh weights were measured immediately after collection. Dry weights were measured after oven drying at 65 °C to a constant weight. The dried materials were ground to a powder, and 0.1 g of the pulverized samples were digested with nitric acid for the determination of Mg^2+^, Ca^2+^, Na^+^, and K^+^ contents. The concentrations of Mg^2+^, Ca^2+^, Na^+^, and K^+^ were determined using a flame atomic absorption spectrophotometer (PinAAcle 900T; PerkinElmer, Inc., Waltham, MA, USA).

### 4.4. Leaf Photosynthetic Index Determination and Defense Enzyme Activity Assays

The net photosynthesis rate (Pn), transpiration rate (Tr), stomatal conductance (gsw), and intercellular CO_2_ concentration (Ci) were measured for the third leaf from the top of each lettuce seedling using an Li-6800 portable photosynthesis system (LI-COR, Lincoln, NE, USA). All measurements were performed between 8:30 a.m. and 12:00 a.m. within 7 d of acetoin treatment. Gas exchange measurements were performed using an artificial light source in the leaf chamber (light intensity, 800 μmol·m^−2^·s^−1^; CO_2_ concentration, 400 μmol·mol^−1^; temperature, 25 °C; relative humidity, 50%).

On days 1, 2, and 3 after acetoin treatment, leaf samples of approximately 0.1 g were collected, frozen in liquid nitrogen for 10 min, and stored at −80 °C until enzyme extraction. The levels of superoxide dismutase (SOD; Cat, No. SOD-1-Y), peroxidase (POD; Cat, No. POD-1-Y), catalase (CAT; Cat, No. CAT-1-Y), malondialdehyde (MDA; Cat, No. MDA-1-Y), and soluble proteins (Cat, No. KMSP-1-W) in the lettuce leaves were measured using antioxidant kits, following the kit instruction manuals (Suzhou Keming Biotechnology Co., Ltd., Suzhou, China).

### 4.5. Preparation of Samples and Transcriptome Sequencing

The hydroponic seedlings were selected for RNA-seq. Uniform seedlings were sprayed with distilled H_2_O (CK group) or 1 mg·mL^−1^ acetoin (AT group). Shoots and roots were sampled 6 and 24 h after treatment, respectively. Each sample group included three biological replicates, and the shoots and roots of the six seedlings in each treatment were combined into one biological replicate. A total of 24 samples (two treatments × two tissues × three biological replicates × two time points) were used for RNA extraction, library preparation, and sequencing.

Total RNA was extracted using a TRIzol reagent kit (Invitrogen, Carlsbad, CA, USA) according to the manufacturer’s instructions. The RNA quality assessment, cDNA preparation, and library sequencing were performed as previously described [40]. Briefly, RNA quality was assessed using an Agilent 2100 Bioanalyzer (Agilent Technologies, Palo Alto, CA, USA) and verified using an RNase-free agarose gel electrophoresis. After extraction of total RNA, the eukaryotic mRNA was enriched using Oligo [43] beads, and prokaryotic mRNA was enriched by removing rRNA via the Ribo-ZeroTM Magnetic Kit (Epicenter, Madison, WI, USA). Next, the enriched mRNA was fragmented using a fragmentation buffer and reverse-transcribed into cDNA with random primers. Second-strand cDNA was synthesized using DNA polymerase I, RNase H, and dNTPs in a buffer. Subsequently, the cDNA fragments were purified using the QiaQuick PCR extraction kit (Qiagen, Venlo, The Netherlands) and end-repaired; poly(A) was added, and the fragments were ligated to Illumina sequencing adapters. The ligation products were size-selected via agarose gel electrophoresis, PCR-amplified, and sequenced using Illumina HiSeq2500 by Gene Denovo Biotechnology Co. (Guangzhou, China).

### 4.6. Screening and Analysis of Differentially Expressed Genes

The DESeq2 package was used to identify differentially expressed genes (DEGs) between groups, and the edgeR package was used to identify DEGs between samples [40]. Genes with a false-discovery rate below 0.05 and an absolute fold change ≥ 2 were considered DEGs, as previously described [40]. The function of DEGs was investigated using Gene Ontology (GO) [13] enrichment and Kyoto Encyclopedia of Genes and Genomes (KEGG) pathway enrichment analyses (https://www.omicsmart.com/).

### 4.7. Quantitative Real-Time PCR

To verify the reliability of the expression trends of DEGs, 10 DEGs from hydroponic samples collected at 6 and 24 h after treatment were randomly selected and verified using quantitative real-time PCR (qRT-PCR) (Table 3). cDNA was synthesized using a PrimeScript™ RT Master Mix kit (Takara Biotech, Dalian, China) according to the manufacturer’s instructions. qRT-PCR amplification was performed using a TB Green^®^ Premix Ex Taq™ II kit (Tli RNaseH Plus, Takara Biotech, Dalian, China) with a CFX96 RT-PCR Detection System (Bio-Rad Company, Pleasanton, CA, USA). *LsGAPC, LsActin,* and *LsTublin* were selected as internal reference genes. The relative expression level of genes was determined using the 2^−ΔΔCt^ method [44], with two technical replicates and three independent biological replicates per gene.

### 4.8. Statistical Analysis

IBM SPSS (version 26.0; IBM Corp., Armonk, NY, USA) for Windows was used for the statistical analyses. All data are expressed as the mean ± standard error. Significant differences were calculated using independent sample *t*-test. One-way analysis of variance was performed to determine differences in the relative expression levels of the genes determined using qRT-PCR. Results were considered significant at a threshold of *p* < 0.05.

## 5. Conclusions

This study investigated the mechanisms underlying plant growth and stress alleviation after acetoin spraying (Figure 10). Acetoin treatment was found to promote shoot and root growth, enhance net leaf photosynthesis, and increase the activity of several defense enzymes in lettuce seedlings grown under saline conditions. In addition, under non-saline conditions, ethylene and ABA signal transduction and MAPK signaling pathway cascades were regulated by acetoin. These pathways are crucial for plants to adapt and respond to various biotic and abiotic stresses, and the modulation by acetoin indicates its potential in fine-tuning the plant’s stress-response mechanisms. The insights gained from this research into the molecular mechanisms of how acetoin impacts plant growth and combats salt stress are not only scientifically significant but also hold substantial promise for agricultural applications. These parsing processes can develop strategies for better bio-fertilizer performance, leading to more resilient and productive crops, especially in saline environments. The prospect of acetoin as a tool for enhancing plant resilience and productivity is particularly exciting given the growing challenges of climate change and the need for sustainable agricultural practices. However, since this study was conducted under laboratory conditions, further testing under field conditions is necessary to verify the potential of acetoin to promote growth and alleviate environmental stress, ensuring its full integration into agricultural practices.

## Figures and Tables

**Figure 1 plants-13-03312-f001:**
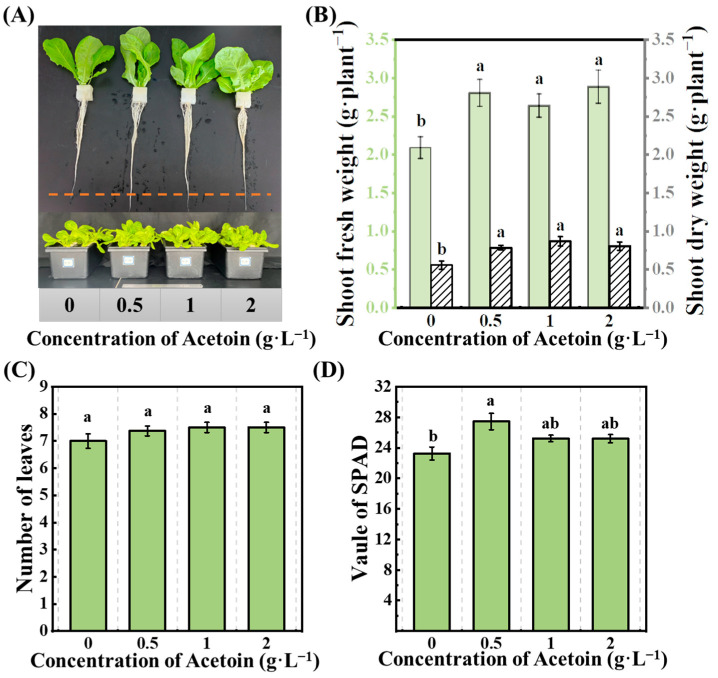
Effects of acetoin (AT) on phenotypic response of lettuce in the hydroponic experiment. (**A**) Phenotype of plants, (**B**) shoot dry weight and root dry weight, (**C**) number of leaves, and (**D**) value of SPAD under CK (non-AT control) and AT treatments. Different letters indicate significant differences (*p* < 0.05) according to an independent sample *t*-test.

**Figure 2 plants-13-03312-f002:**
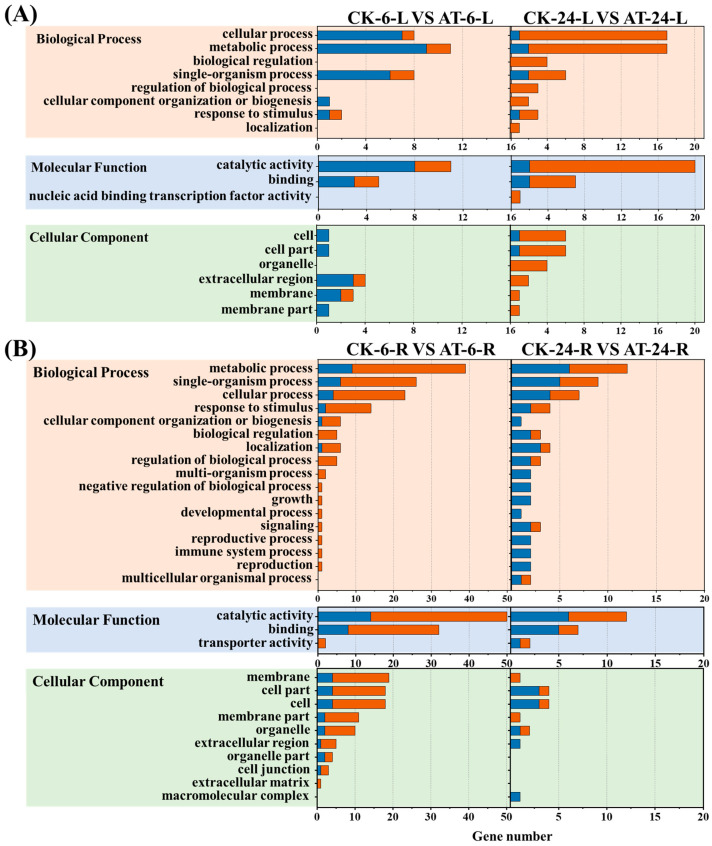
Numbers of differentially expressed genes associated with biological processes, cellular components, and molecular functions found using Gene Ontology [13] enrichment analysis. (**A**) GO analysis of DEGs in the shoots of Lettuce. (**B**) GO analysis of DEGs in the roots of Lettuce. Blue bar represents up-regulated DEGs. Orange bar represents down-regulated DEGs. CK6/24 represents the non-acetoin (AT) control at 6 h (24 h post treatment); AT6/24 represents the AT control at 6 h (24 h post treatment). L and R represent the shoots and roots of lettuce seedlings, respectively.

**Figure 3 plants-13-03312-f003:**
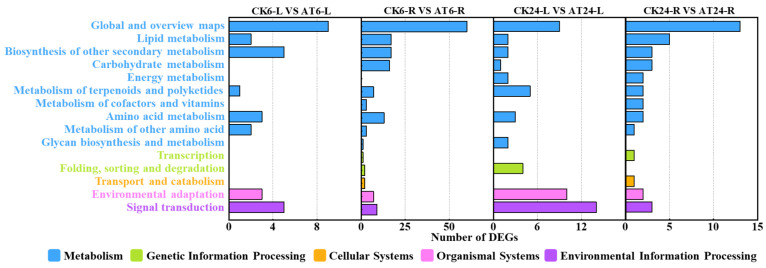
Kyoto Encyclopedia of Genes and Genomes (KEGG) classification of differentially expressed genes (DEGs). Results are summarized as five main KEGG A Classes: metabolism, genetic information processing, cellular processes, organismal systems, and environmental information processing. CK6/24 represents the non- acetoin (AT) control at 6 h (24 h post treatment); AT6/24 represents the AT control at 6 h (24 h post treatment). L and R represent the shoots and roots of lettuce seedlings, respectively. Numbers indicate the number of DEGs.

**Figure 4 plants-13-03312-f004:**
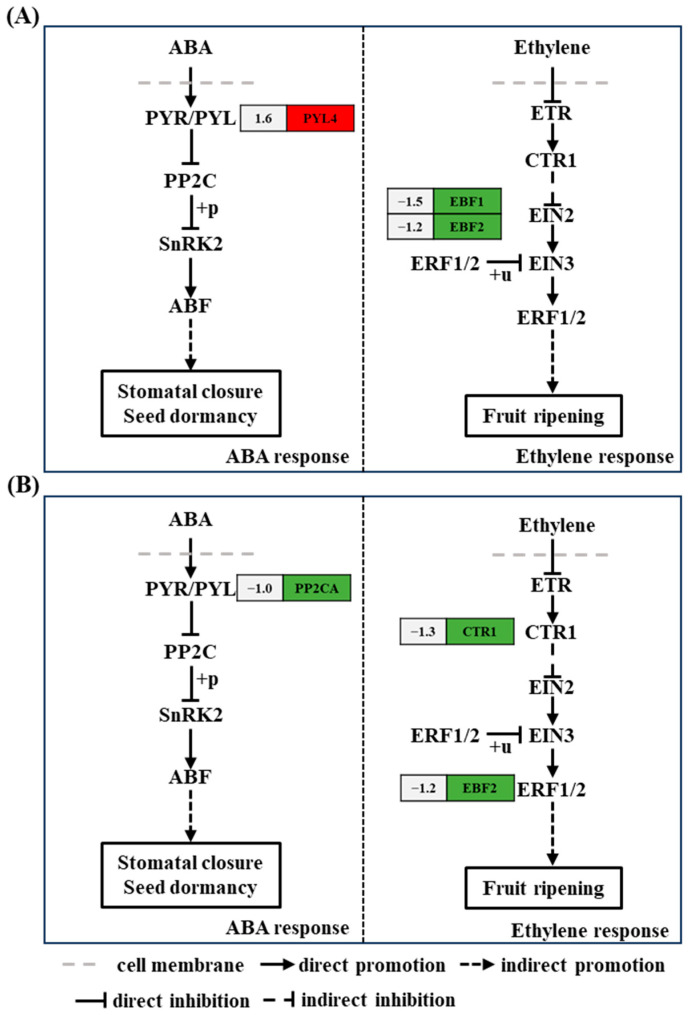
Kyoto Encyclopedia of Genes and Genomes (KEGG) pathway analysis of differentially expressed genes (DEGs) related to plant hormones and signal transduction at 6 and 24 h post acetoin treatment. (**A**) KEGG pathway analysis of DEGs related to plant hormones and signal transduction at 6 h. (**B**) and at 24 h. Red and green indicate up- and downregulated genes, respectively and the numbers on the left represent the differential expression of genes. ABA represents abscisic acid; *PYR/PYL*, *PP2C*, *SnRK2*, *ABF*, *ETR*, *CTR1*, *EIN2*, *EIN3*, and *ERF1/2* represent the genes encoding abscisic acid receptor PYR/PYL family, protein phosphatases 2C, SNF1-related type 2 protein kinase, ABA-response element binding factors, ethylene receptor, serine/threonine-protein kinase CTR1, ethylene-insensitive protein 2, ethylene-insensitive protein 3, and ethylene-responsive transcription factor 1/2, respectively.

**Figure 5 plants-13-03312-f005:**
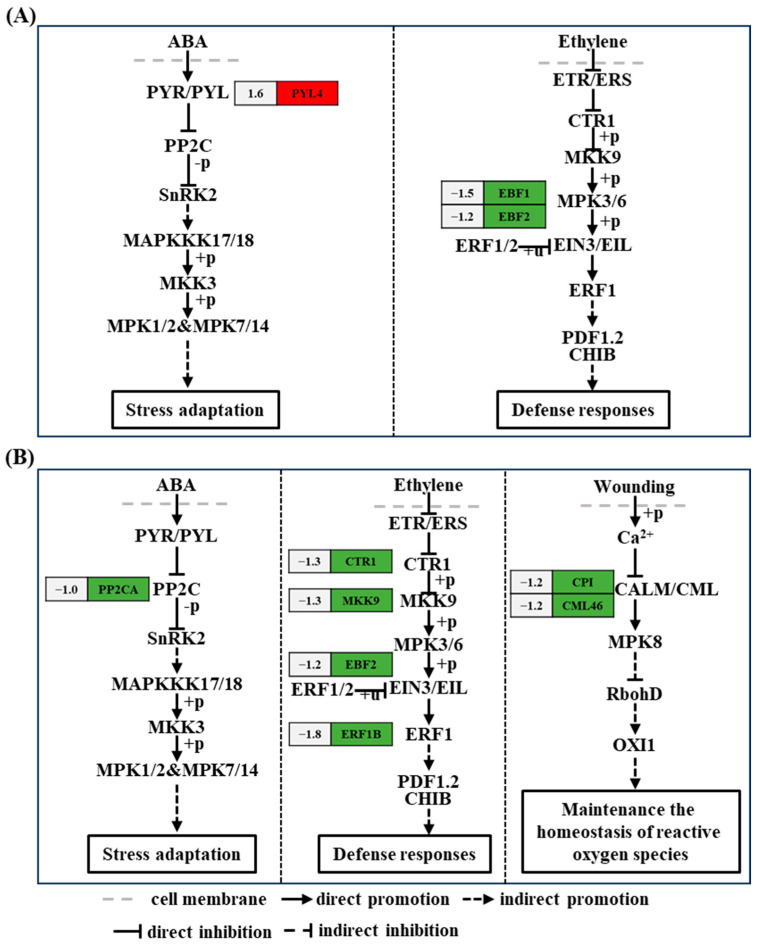
Kyoto Encyclopedia of Genes and Genomes (KEGG) pathway analysis of differentially expressed genes (DEGs) related to MAPK signaling pathway–plant at 6 and 24 h post acetoin treatment. KEGG pathway analysis of DEGs related to MAPK signaling pathway–plant at 6 (**A**) and 24 h (**B**). Red and green indicate up- and downregulated genes, respectively and the numbers on the left represent the differential expression of genes. ABA represents abscisic acid; *PYR/PYL*, *PP2C*, *SnRK2*, *MAPKKK17/18*, *MKK3/9*, *ETR/ERS*, *CTR1*, *EIN3/EIL*, *ERF1*, *PDF1.2*, *CHIB*, *CALM/CML*, *RbohD*, and *OXI1* represent the genes encoding abscisic acid receptor PYR/PYL family, protein phosphatases 2C, SNF1-related type 2 protein kinase, mitogen-activated protein kinase kinase kinase 17/18, mitogen-activated protein kinase 3/9, ethylene receptor, serine/threonine-protein kinase CTR1, ethylene-insensitive protein 3, ethylene-responsive transcription factor 1, defensin-like protein 16, basic endochitinase B, calmodulin/calmodulin-like protein, respiratory burst oxidase homologous protein D, and serine/threonine-protein kinase OXI1, respectively. MPK1/2, MPK7/14, MPK3/6, and MPK8 represents the genes encoding mitogen-activated protein kinase 1/2, 7/14, 3/6, and 8, respectively.

**Figure 6 plants-13-03312-f006:**
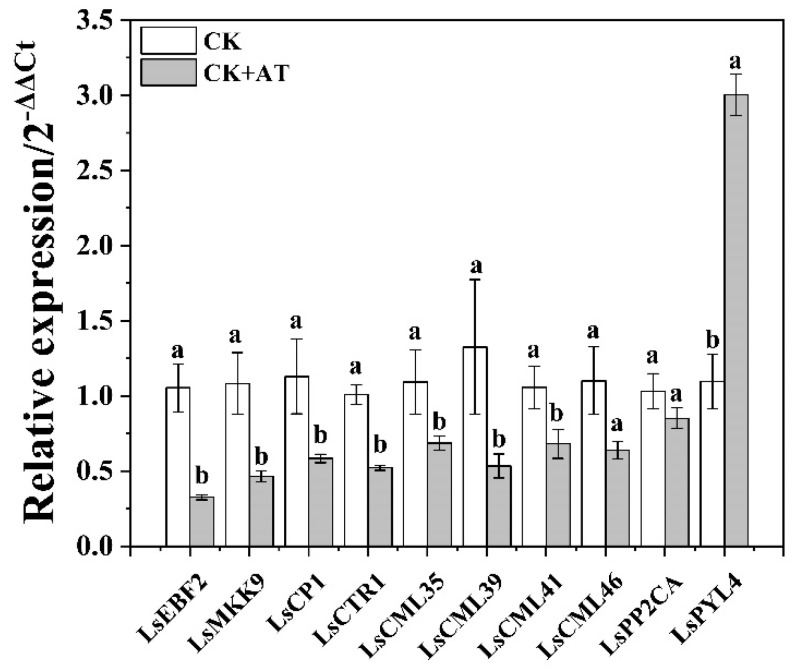
Relative expression levels of 10 differentially expressed genes were measured using qRT-PCR (2^−ΔΔCt^). CK represents the non-acetoin (AT) control. According to independent sample *t*-test, different letters indicate significant differences (*p* < 0.05). Ls represents lettuce; *EBF2*, *MKK9*, *CP1*, *CTR1*, *CML35/39/41/46*, *PP2CA*, and *PYL4* represent the genes encoding the EIN3-binding F-box protein 2, mitogen-activated protein kinase 9, calmodulin, serine/threonine-protein kinase, calmodulin-like protein 35/39/41/46, protein phosphatases 2C, and ABA receptor PYR1-like protein, respectively.

**Figure 7 plants-13-03312-f007:**
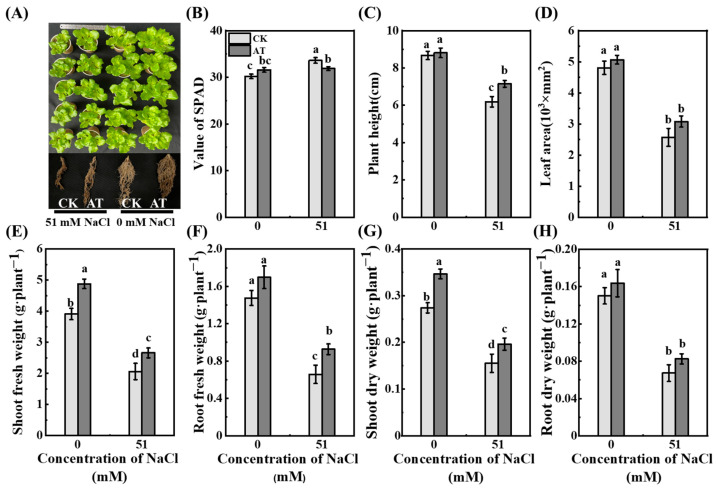
Effects of acetoin (AT) on the growth of lettuce under non-saline and saline conditions in the pot experiment. (**A**) Phenotype of shoots, (**B**) value of SPAD, (**C**) plant height, (**D**) leaf area, (**E**) shoot fresh weight, (**F**) root fresh weight, (**G**) shoot dry weight, and (**H**) root dry weight under CK (non-AT control) and AT treatments. According to independent sample *t*-test, different letters indicate significant differences (*p* < 0.05).

**Figure 8 plants-13-03312-f008:**
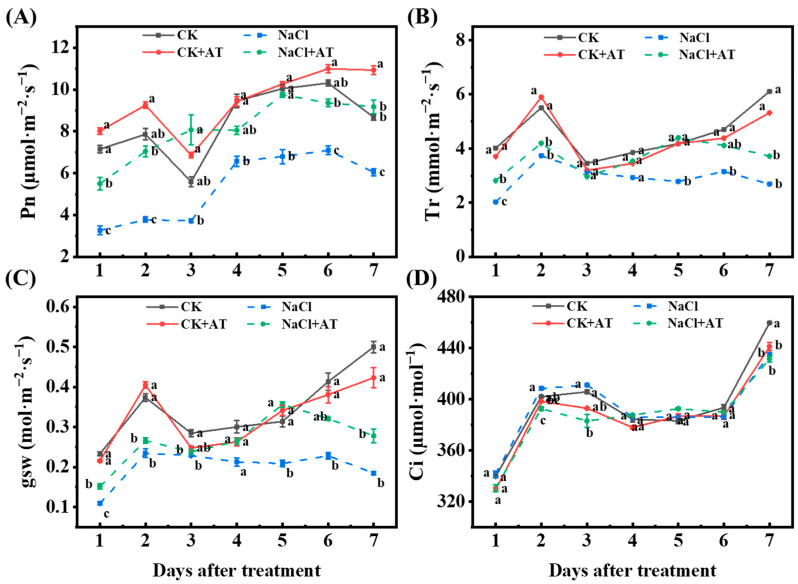
Effects of acetoin (AT) on the photosynthetic parameters of lettuce seedlings under non-saline and saline conditions in the pot experiment. (**A**) Net photosynthetic rate (Pn), (**B**) transpiration rate (Tr), (**C**) stomatal conductance (gsw), and (**D**) intercellular CO_2_ concentration (Ci) under CK (non-AT and non-NaCl treatment control), CK + AT, NaCl (3 g NaCl per kg of soil), and NaCl + AT treatments. According to independent sample *t*-test, different letters indicate significant differences (*p* < 0.05).

**Figure 9 plants-13-03312-f009:**
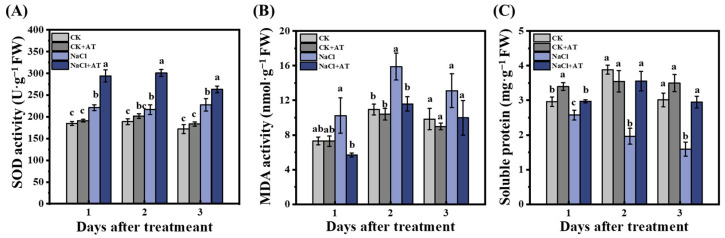
Dynamic changes in the defense enzyme activities in leaves of lettuce seedlings under non-saline and saline conditions in the pot experiment. (**A**) Superoxide dismutase (SOD) activity; (**B**) malondialdehyde (MDA) activity; and (**C**) soluble protein content under CK (non-AT and non-NaCl treatment control), CK + AT, NaCl (3 g NaCl per kg of soil), and NaCl + AT treatments. According to independent sample *t*-test, different letters indicate significant differences (*p* < 0.05).

**Figure 10 plants-13-03312-f010:**
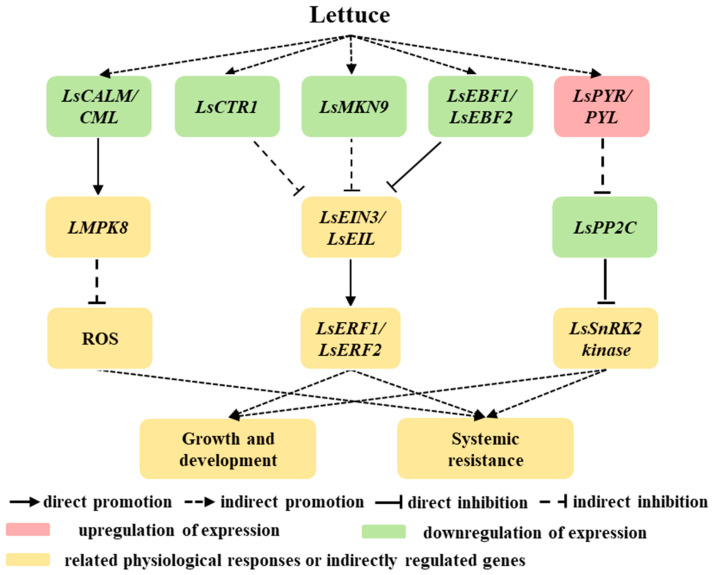
Schematic of growth-promotion mechanisms in lettuce influenced by acetoin. Red modules represent upregulated genes or positive promotion. Green modules represent downregulated genes. Yellow modules represents related physiological responses or indirectly regulated genes. Ls represents lettuce; *CALM/CML*, *CTR1*, *MKK9*, *EBF1/2*, *PYR/PYL*, *MPK8*, *EIN3/EIL*, *PP2C*, *ERF1/2*, and *SnRK2* represent the gene encoding calmodulin/calmodulin-like protein, serine/threonine-protein kinase CTR1, mitogen-activated protein kinase kinase 9, EIN3-binding F-box protein 1/2, abscisic acid receptor PYR/PYL family, mitogen-activated protein kinase 8, ethylene-insensitive protein 3, protein phosphatases 2C, ethylene-responsive transcription factor 1/2, and SNF1-related type 2 protein kinase, respectively; ROS represents reactive oxygen species.

**Table 1 plants-13-03312-t001:** Kyoto Encyclopedia of Genes and Genomes enrichment pathways and significantly differentially expressed genes (Q value ≤ 0.05) under acetoin (AT) treatment in lettuce.

	Pathway ID	Pathway	Numbers of Significant Genes	Q Value
CK6-L-vs-AT6-L	ko00941	Flavonoid biosynthesis	3	1.46 × 10^−3^
	ko04075	Plant hormone signal transduction	5	2.19 × 10^−3^
	ko04016	MAPK signaling pathway–plant	3	2.25 × 10^−2^
CK24-L-vs-AT24-L	ko04016	MAPK signaling pathway–plant	8	9.31 × 10^−5^
	ko04075	Plant hormone signal transduction	9	4.68 × 10^−4^
	ko04626	Plant–pathogen interaction	8	6.48 × 10^−3^
CK6-R-vs-AT6-R	ko01110	Biosynthesis of secondary metabolites	44	1.20 × 10^−7^
	ko00940	Phenylpropanoid biosynthesis	16	7.65 × 10^−7^
	ko00071	Fatty acid degradation	7	6.97 × 10^−4^
	ko01100	Metabolic pathways	51	4.12 × 10^−3^
	ko00350	Tyrosine metabolism	6	4.21 × 10^−3^
	ko01040	Biosynthesis of unsaturated fatty acids	5	5.74 × 10^−3^

Note: CK6 or CK24 represents the non-AT control at 6 or 24 h post treatment; AT6 or AT24 represents the AT control at 6 or 24 h post treatment. L and R represent the shoots and roots of lettuce seedlings, respectively.

**Table 2 plants-13-03312-t002:** Variation in ionic elements on lettuce sprayed by acetoin for 21 d after transplantation under non-saline and saline conditions in the pot experiment.

Organ	Treatment	Mg^2+^	Ca^2+^	Na^+^	K^+^	K^+^/Na^+^
		(mg·g^−1^ DW)	(mg·g^−1^ DW)	(mg·g^−1^ DW)	(mg·g^−1^ DW)	
Shoot	CK	5.27 ± 0.09 b	3.57 ± 0.46 ab	6.07 ± 0.12 c	9.48 ± 0.42 c	1.56 ± 0.04 b
	CK + AT	5.72 ± 0.21 a	3.93 ± 0.24 ab	5.98 ± 0.10 c	10.31 ± 0.42 c	1.73 ± 0.07 ab
	NaCl	4.35 ± 0.13 c	2.87 ± 0.44 b	9.07 ± 0.15 a	12.46 ± 0.41 b	1.38 ± 0.05 c
	NaCl + AT	3.96 ± 0.15 c	4.30 ± 0.57 a	7.85 ± 0.19 b	14.10 ± 0.75 a	1.79 ± 0.06 a
Root	CK	5.74 ± 0.27 a	1.44 ± 0.29 c	6.38 ± 0.16 b	2.41 ± 0.13 b	0.38 ± 0.01 b
	CK + AT	5.84 ± 0.17 a	3.70 ± 0.18 a	6.26 ± 0.20 b	2.03 ± 0.13 b	0.33 ± 0.03 b
	NaCl	4.10 ± 0.16 b	0.73 ± 0.05 c	8.40 ± 0.25 a	5.44 ± 0.33 a	0.65 ± 0.04 a
	NaCl + AT	3.94 ± 0.13 b	2.45 ± 0.31 b	6.65 ± 0.64 b	5.18 ± 0.46 a	0.82 ± 0.11 a

Note: CK represents the non-acetoin (AT) and non-NaCl treatment control; NaCl represents the treatment with 3 g NaCl per kg of soil and non-AT treatment control; and DW represents dry weight. Data are presented as means ± standard error, n = 10. According to independent sample *t*-test, different letters indicate significant differences (*p* < 0.05).

**Table 3 plants-13-03312-t003:** Primer sequences used for qRT-PCR.

Gene Name	Sequence of Forward Primer (5′–3′)	Sequence of Reverse Primer (5′–3′)	Accession Number	Amplified Product Size (bp)
*PYL4*	CAAGAGCTGCAACGTGATTCTC	TTGTTTTGTCGCAGTTTGGTGA	XM_023876294	218
*EBF2*	GCGGAACCCTAGAAGTCTTGAA	AAGAACGGCGTACACTTGTTTG	XM_023917299	223
*MKK9*	GCGATATCTGGAGCTTGGGG	GATGCTGTCCACCTCTTGCT	XM_023910820	211
*CP1*	CGATGATGACGGTTGCTGATTC	TCTCCATACCCTACCTTTCCGT	XM_023909929	163
*CTR1*	TCAAATTCCCGACGAGCACA	GCATAGCTACTCCCACTCGA	XM_023902326	219
*CML35*	CTCTGCATCCTCTCCCTCTTTC	TTCATGTGAGTGGGTCGGTAAG	XM_023885715	225
*CML39*	AGATGATGAAGGAAGGGGAGGA	GCCATTAACGTCGAATCTTCCG	XM_023889008	170
*CML41*	CCGTGGAAAACCTGCTTGAAAT	CTGGAGTTCTTGTAAGTGGGCT	XM_023900613	232
*CML46*	TTCCCCGGAACCCTACATGT	TCCAATCTCGGTTGCTCCTC	XM_023915611	246
*PP2CA*	ATGCCTCCACCTGACAGTAGTA	TCCAACACATCCTCTCCCAAAG	XM_023906444	212
*ACTIN*	ATGGCCGACACTGAGGATATTC	ACATAAGCATCTTTCTGGCCCA	XM_023878805	167
*GAPC*	GTTGTTGGTGTGAACGAGAAGG	ACCTCTCCAGTCCTTAGCAGAT	XM_023875356	201
*TUBLIN*	CAGATGCCCAGTGACAAAACAG	AGGTCGACGATTTCTTTTCCGA	XM_023889059	248

## Data Availability

The datasets generated during and/or analyzed during the current study are available from the corresponding author on reasonable request.

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
