# Peer review of "Acetoin Promotes Plant Growth and Alleviates Saline Stress by Activating Metabolic Pathways in Lettuce Seedlings"

_plants, 2024, doi:10.3390/plants13233312_

Round 1
Reviewer 1 Report
Comments and Suggestions for Authors
The manuscript entitled, "Acetoin promotes plant growth and alleviates saline stress in lettuce seedlings" is a good study.
In materials and methods, authors have mentioned two levels of salinity (0 and 2%), but in all figures and tables, I cannot see anything related to normal and saline conditions. Add the data of rice under control conditions in comparison with saline conditions. However, there are some comments which can improve this article.
Title
· It should be revised, it should not only relate to growth only
Abstract
· Mention the treatments such as acetoin level
· Results were poorly explained, need to revise them and write in a mechanistic way.
· Remove the heads like background and aims, materials and method etc.
Introduction
· Write the saline area in the World and in China. Explain, Why is it increasing every year?
· Add a paragraph on how the salinity effect lettuce growth and gene regulations
· Overall, try to cite new studies and remove old.
Materials and Methods
· By following which protocol authors have measured them “The physicochemical characteristics of the soil were as follows: 11.0 g kg-1 organic matter, 29.7 mg kg-1 alkali-hydrolyzed nitrogen, 68.2 mg kg-1 available P, 271.2 mg kg-1, available K, and a pH of 7.08”.
· 3 g NaCl kg-1 soil? Why have you chosen this level? And write this in mM or dS m-1.
· EC can not be zero, if you have normal soil, there should be some EC of that soil, mention the EC as well as a control.
· How have authors imposed the salinity?
Results
· Results are well written
Discussion
· Overall, the discussion is very poorly written. Need to add more and clearer studies, on how the plant growth and other parameters affected by salinity and Acetoin improve them.
References
· In text, authors mentioned the authors name but in the bibliography mentioned the numbering. So, improve it according to journal requirements.
Comments on the Quality of English LanguageMinor editing is needed
Author Response
Comments 1: The manuscript entitled, "Acetoin promotes plant growth and alleviates saline stress in lettuce seedlings" is a good study. In materials and methods, authors have mentioned two levels of salinity (0 and 2%), but in all figures and tables, I cannot see anything related to normal and saline conditions. Add the data of rice under control conditions in comparison with saline conditions. However, there are some comments which can improve this article.
Response 1: Thank you very much for your affirmation. We agree with this comment. We have designed two levels of salinity (0 and 3%) in the Figure 7 and added the annotation related to normal and saline conditions for easier understanding in the revised manuscript.
Comments 2: In the Title, it should be revised, it should not only relate to growth only.
response 2: Thank you for your suggestion. We agree with this comment. Therefore, we have revised the title as follows: “Acetoin promotes plant growth and alleviates saline stress through activating metabolic pathways in lettuce seedlings”.
Comments 3: In the Abstract, mention the treatments such as acetoin level.
response 3: Thank you for this valuable suggestion. We have added treatments of the acetoin level in the abstract of revised manuscript. This change can be found in line 12-15 as follow: "Lettuce (Lactuca sativa L. var. ramosa Hort.) seedlings in hydronics and pots added to non-saline and saline soil were foliar-sprayed with 10 ml of 0 or 1 mg·mL-1 acetoin at 7 and 14 d after trans-plantation and harvested 7 d after the second spray.”
Comments 4: In the Abstract, results were poorly explained, need to revise them and write in a mechanistic way.
Response 4: Thank you for your insightful comment. We have revised the results and write in mechanistic way in the abstract of the revised manuscript. This change can be found in line 18-27 as follow: “The transcriptomic analysis revealed acetoin application resulted in 177 differentially expressed genes (DEGs; 39 upregulated and 138 downregulated) in shoots and 397 DEGs (112 upregulated and 285 downregulated) in roots. These DEGs, mainly involved in plant hormone signal transduction and the mitogen-activated protein kinase (MAPK), have potential to trigger plants responses to various environmental stimuli, including stress and developmental signals. Under saline conditions, acetoin-treated plants showed increased net leaf photosynthesis and activities of several defense enzymes, indicating that acetoin enhances both fundamental growth and the plant's stress defenses, especially against salinity. In summary, acetoin appears to act through a complex interplay of genetic and biochemical mechanisms, influencing key signaling pathways and physiological processes that lead to improved growth and stress tolerance in lettuce seedlings.”
Comments 5: In the Abstract, remove the heads like background and aims, materials and method etc.
Response 5: Thank you for pointing this out. Accordingly, we have removed the heads like background and aims, materials and method in the abstract.
Comments 6: In the Introduction, write the saline area in the World and in China. Explain, Why is it increasing every year?
Response 6: Thank you for your query. Accordingly, we have provided information on the saline area around the world and have listed the reasons for the increasing area in the line 35-40 of the revised introduction of manuscript. The revised text is presented as follow: "Among them, soli salinity increase is one of the main challenges of contemporary agriculture. The global area of salt-affected soils is approximately 1.0×109 hectares, accounting for about 25% of the Earth's land area [5]. Human activities such as irrigation with saline water, deforestation, and improper agricultural practices with the excessive use of chemical fertilizers, exacerbate soil salinity [6]."
Comments 7: In the Introduction, Add a paragraph on how the salinity effect lettuce growth and gene regulations.
Response 7: Thank you for pointing this out. We have added a paragraph to describe the salinity effect growth and gene regulations about pathways in the introduction. Additionally, we have combined the information on the selection of lettuce as model species in this paragraph. The revised text is presented in line 58-67 as follow: “Lettuce (Lactuca sativa L. var. ramosa Hort.), revered for its rich nutritional profile and diverse health benefits, is one of the most important vegetables worldwide, but its moderate sensitivity to salinity seriously restricts yield [4,15]. Previous studies have proved that the plant physiology was impaired by osmotic stress response to salinity, resulting in stomatal closure to decrease water loss by transpiration [16]. Furthermore, photosynthesis is affected by Na+ toxicity, where a high cytosolic accumulation of Na+ tends to replace some K+ and consequently vital enzymatic activities involving K+ are disrupted [17]. Additionally, salinity has been demonstrated to induce the modulation of endogenous phytohormones levels, which in turn affect the signaling pathways (e.g., MAPK signaling pathway) involved in the downstream changes in roots, leaves and cellular structures [18].”
Comments 8: In the Introduction, Overall, try to cite new studies and remove old.
Response 8: Thank you for your suggestions. We have cited new studies and removed old refs.
Comments 9: In the Materials and Methods, by following which protocol authors have measured them “The physicochemical characteristics of the soil were as follows: 11.0 g kg-1 organic matter, 29.7 mg kg-1 alkali-hydrolyzed nitrogen, 68.2 mg kg-1 available P, 271.2 mg kg-1, available K, and a pH of 7.08”.
Response 9: Thank you for pointing this out. The physicochemical characteristics of the soil have been reported in prior research, so the protocol is not extensively detailed here. We have added this reference in the revised manuscript. This change can be found in line 87-90 as follow: " The physicochemical characteristics of the soil were reported in prior research [20], as detailed below: 11.0 g kg-1 organic matter, 29.7 mg kg-1 alkali-hydrolyzed nitrogen, 68.2 mg kg-1 available P, 271.2 mg kg-1 available K, and a pH of 7.08. "
Comments 10: In the Materials and Methods, 3 g NaCl kg-1 soil? Why have you chosen this level? And write this in mM or dS m-1.
Response 10: Thank you for your queries and suggestion. The 3 g NaCl kg-1 soil represents that 3 grams of NaCl are mixed with every kilogram of fresh sieved soil. The 3 grams of NaCl is equal with 51 mmol of NaCl. Then, the effects of NaCl level have been explored in our lab. We set up a gradient of salt concentrations at 2, 3, and 4 g NaCl kg-1 soil. High salt concentrations (4 g NaCl kg-1) resulted in seedling death, which is not conducive to the normal growth of lettuce, while low salt concentrations (2 g NaCl kg-1) led to significant growth fluctuations in lettuce seedlings with a relatively minor salt inhibition effect. Lettuce is inherently a moderately salt-sensitive plant, and the level of 3 g NaCl kg-1 soil meets the requirement for lettuce survival and growth while exerting a certain level of inhibition.
Comments 11: In the Materials and Methods, EC can not be zero, if you have normal soil, there should be some EC of that soil, mention the EC as well as a control.
Response 11: Thank you for your suggestion. We agree with this comment. Electrical conductivity (EC) in our original soils (the controls) can’t be zero. However, we have not discussed EC in this study and the involved saline treatment focused on the addition of exogenous sodium chloride (NaCl), including 0 g or 3 g per kg of fresh weight soil.
Comments 12: In the Materials and Methods, how have authors imposed the salinity?
Response 12: Thank you for your query. We have imposed the salinity in Materials and Methods of the manuscript as follow: The saline soil was obtained by adding 51 mM of NaCl, and mixed thoroughly. This change can be found as follow: "The saline soil was obtained by adding 51 mM of NaCl, and mixed thoroughly."
Comments 13: Results are well written.
Response 13: Thank you very much for your affirmation.
Comments 14: Overall, the discussion is very poorly written. Need to add more and clearer studies, on how the plant growth and other parameters affected by salinity and Acetoin improve them.
Response 14: Thank you for your suggestions. We agree with this comment. Accordingly, we have added some clearer studies and replaced some old studies in the revised discussion, to better explain the plant growth and other parameters affected by salinity and acetoin.
Comments 15: In text, authors mentioned the authors name but in the bibliography mentioned the numbering. So, improve it according to journal requirements.
Response 15: Thank you for pointing this out. We have improved the numbering of references according to the journal guidelines in the text.
Reviewer 2 Report
Comments and Suggestions for Authors
Dear authors, i read with interest your article. My comments are reported in the attached PDF, please follow it.
I think that after this modification the article will be suitable for the publication in Plants.
Results very clear

Author Response
Comments 1: I think that after this modification the article will be suitable for the publication in Plants. Results very clear.
Reponse 1: Thank you very much for your affirmation. As suggested by you, the responses to the comments have been addressed point-to-point in the PDF of revised manuscript.

Reviewer 3 Report
Comments and Suggestions for Authors
The manuscripts presents innovative experiment on the use of promising stress-releasing acetoin in plant production. The research is consistent with present needs of sustainable crop production under stress conditions. The approach of the authors to the problem is detailed, based on the various levels of plant response to the acetoin: it treats about plant growth, many issues of plant physiology and scrutinized expression of genes.. Unfortunately, it also includes some deficiencies, like the most important scarce number of plant material – only 10 seedlings/treatment. Yet, in appreciation of its very impressive and comprehensive procedure I would recommend it to be published if the following shortages are complemented
1. English requires editing. The most common ubiquitous error was wrong choice of verbs, then wrong prepositions
2. In Results, both the text and graphs/figures do not not specify if there are the results of hydroponic or pot experiment
3. Discussion is quite superficial and based on the small selection od references (only 38).
4. Conclusion is too superficial. It should be more related to future prospects than to the form of Abstract.
More remarks are given in the text.

1. English requires editing. The most common ubiquitous error was wrong choice of verbs, then wrong prepositions.
Author Response
Comments 1: The manuscripts present innovative experiment on the use of promising stress-releasing acetoin in plant production. The research is consistent with present needs of sustainable crop production under stress conditions. The approach of the authors to the problem is detailed, based on the various levels of plant response to the acetoin: it treats about plant growth, many issues of plant physiology and scrutinized expression of genes. Unfortunately, it also includes some deficiencies, like the most important scarce number of plant material – only 10 seedlings/treatment. Yet, in appreciation of its very impressive and comprehensive procedure I would recommend it to be published if the following shortages are complemented.
Reponse 1: Thank you very much for your affirmation. As suggested for you, we have revised the deficiencies in the manuscript and provided corresponding responses to the comments as follow.
Comments 2: English requires editing. The most common ubiquitous error was wrong choice of verbs, then wrong prepositions.
Response 2: Thank you for your constructive comment. As suggested by you, we have revised the English editing in the revised manuscript. The point-to-point responses have been reviewed in the PDF file of the manuscript.
Comments 3: In Results, both the text and graphs/figures do not not specify if there are the results of hydroponic or pot experiment.
Response 3: Thank you for pointing this out. Accordingly, we have added the corresponding cultivation conditions (hydroponic or pot experiment) in the text and the annotations of graphs/figures. We added "in the hydroponic experiment" to the annotation of Figure 1, the line 181 and line 120 of the text. We added "in the pot experiment" to the annotations of Figure 7-9 and Table 3, as well as the line 319 of the text.
Comments 4: Discussion is quite superficial and based on the small selection od references (only 38).
Response 4: Thank you for your suggestion. We have added some clearer studies up to 46 in the revised manuscript.
Comment 5: Conclusion is too superficial. It should be more related to future prospects than to the form of Abstract.
Response 5: Thank you for your suggestion. Accordingly, we have rewritten parts of the conclusion to better elaborate on the future prospects of acetoin as follow:
Conclusion
This study investigated the mechanisms underlying plant growth and stress alleviation after acetoin spraying (Fig. 10). Acetoin treatment was found to promote shoot and root growth, enhance net leaf photosynthesis, and increase the activity of several defense enzymes in lettuce seedlings grown under saline conditions. In addition, under non-saline conditions, ethylene and ABA signal transduction and MAPK signaling pathway cascades were regulated by acetoin. These pathways are crucial for plants to adapt and respond to various biotic and abiotic stresses, and the modulation by acetoin indicates its potential in fine-tuning the plant's stress response mechanisms. The insights gained from this research into the molecular mechanisms of how acetoin impacts plant growth and combats salt stress are not only scientifically significant but also hold substantial promise for agricultural applications. These parsing processes can develop strategies for better bio-fertilizer performance, leading to more resilient and productive crops, especially in saline environments. The prospect of acetoin as a tool for enhancing plant resilience and productivity is particularly exciting given the growing challenges of climate change and the need for sustainable agricultural practices. However, since this study was conducted under laboratory conditions, further testing under field conditions is necessary to verify the potential of acetoin to promote growth and alleviate environmental stress, ensuring its full integration into agricultural practices.

Round 2
Reviewer 1 Report
Comments and Suggestions for Authors
Manuscript Title: Acetoin promotes plant growth and alleviates saline stress through activating metabolic pathways in lettuce seedlings
Authors have addressed all the comments according to suggestion. Now, it is recommended for publication.
Comments on the Quality of English LanguageMinor editing of English language required.